# Purification and characterization of antifungal lipopeptide produced by *Bacillus velezensis* isolated from raw honey

**Zirui Ray Xiong** [1]*, **Mario Cobo**[1], **Randy M. Whittal**[2], **Abigail B. Snyder**[1], **Randy W. Worobo**[1]

**1** Department of Food Science, Cornell University, Ithaca, NY, United States of America, **2** Department of Chemistry, University of Alberta, Edmonton, AB, Canada

* zx97@cornell.edu

**Data Availability Statement:** All relevant data are within the manuscript and its Supporting Information files.

## Abstract

Raw honey contains a diverse microbiota originating from honeybees, plants, and soil. Some gram-positive bacteria isolated from raw honey are known for their ability to produce secondary metabolites that have the potential to be exploited as antimicrobial agents. Currently, there is a high demand for natural, broad-spectrum, and eco-friendly bio-fungicides in the food industry. Naturally occurring antifungal products from food-isolated bacteria are ideal candidates for agricultural applications. To obtain novel antifungals from natural sources, we isolated bacteria from raw clover and orange blossom honey to evaluate their antifungal-producing potential. Two *Bacillus velezensis* isolates showed strong antifungal activity against food-isolated fungal strains. Antifungal compound production was optimized by adjusting the growth conditions of these bacterial isolates. Extracellular proteinaceous compounds were purified via ammonium sulfate precipitation, solid phase extraction, and RP-HPLC. Antifungal activity of purified products was confirmed by deferred overlay inhibition assay. Mass spectrometry (MS) was performed to determine the molecular weight of the isolated compounds. Whole genome sequencing (WGS) was conducted to predict secondary metabolite gene clusters encoded by the two antifungal-producing strains. Using MS and WGS data, we determined that the main antifungal compound produced by these two *Bacillus velezensis* isolates was iturin A, a lipopeptide exhibiting broad spectrum antifungal activity.

## Introduction

Antifungal resistance in medically and agriculturally relevant fungi is increasing globally, straining the limited selection of safe and effective antifungal agents. The development of novel antifungal agents is much slower than the spread of antifungal resistant strains, which presents a serious human health and food security problem [1]. In the medical field, fungal infections are extremely difficult to treat. Fungicides that are broad spectrum, effective, and safe to use, are limited. Furthermore, the prevalence of multi-drug resistant fungal pathogens

**Funding:** This study was supported by the U.S. Department of Agriculture, National Institute of Food and Agriculture multistate project S-1077, and the College of Agriculture and Life Sciences at Cornell University. The funders had no role in study design, data collection and analysis, decision to publish, or preparation of the manuscript.

**Competing interests:** The authors have declared that no competing interests exist.

has been increasing in hospitals and nursing homes [2]. For the widely deployed azole family, resistance has been observed in common fungal pathogens [1]. For example, multi-azole-resistant strains of the opportunistic pathogen *Aspergillus fumigatus* have been isolated from patients with invasive aspergillosis [3]. Fluconazole-resistant *Candida glabrata* with increased resistance to the other first-line antifungal drug echinocandin was also observed, which further limited the available options to treat this infection [4]. Additionally, multidrug-resistant *Candida auris*, first isolated in 2009, has invasively infected patients worldwide through hospital-acquired transmission [5]. In the agricultural field, fungal plant pathogens have also acquired resistance against antifungal agents. Even though more fungicides are available for field application, the rapid rate of antifungal resistance development is alarming. A classic example of an organism with high risk of antifungal resistance development is *Botrytis cinerea*, which is able to adapt to new fungicide classes. Multidrug-resistant *B. cinerea* strains have been isolated in strawberry fields around the world [6]. The predominant class of chemicals used for antifungal treatment of crops is azoles. Scientists have urged to restrict the use of azoles in agriculture, as resistant fungal strains are being continuously isolated from environmental and clinical settings at an increasing rate [7]. However, due to the lack of alternatives, it is still being widely used in economically important crops to avoid crop losses. In contemporary food systems, spoilage caused by fungi is no less serious. Food loss due to fungal spoilage was estimated to account for 5–10% of the world food supply, and post-harvest spoilage was estimated to contribute to 25% of global food waste [8, 9]. In a survey of 51 juice manufacturers, 92% reported experiencing yeast or mold spoilage in their finished product and 89% reported previous occurrences of yeast or mold spoilage of their ingredients [10]. Spoilage fungi are difficult to control due to their ability to survive extreme conditions, like low water activity, limited nutrients, high acidity, and extreme heat treatment. Moreover, the trade-off of common fungal-controlling approaches in the food industry is the negative environmental impact, such as food waste, unsustainable packaging, and environmental damage by synthesized chemicals [11]. Natural bio-fungicide could be a beneficial addition to traditional fungal-controlling approaches and mitigate the environmental impact.

The urgent need for natural, novel, safe, and potent antifungal compounds lead us to seek solutions from natural products, like honey. Raw honey is inhibitory to fungi, partially due to its high sugar content and low water activity [12]. However, a survey comparing the antifungal effects of raw monofloral honey with synthetic honey demonstrated that heather and lavender honey exhibited higher antifungal activity than sugar-based synthetic honey [13, 14]. Other than osmotic inhibition, some chemical components in raw honey are also antifungal: hydrogen peroxide, flavonoids, phenolic acids, lysozymes, and other antioxidant compounds [15]. Additionally, antifungal bacteria are present in raw honey. In previous studies, *Bacillus* spp. strains isolated from raw honey were able to produce a variety of secondary metabolites to inhibit the growth of other microorganisms and gain survival advantages. *B. subtilis* H215 was isolated from raw honey and it was inhibitory to *Byssochlamys fulva* H25 [16]. Another isolate found in US domestic honey, *B. thuringiensis* SF361, showed broad spectrum antifungal activity against *Aspergillus*, *Penicillium*, *Byssochlamys*, and *Candida albicans* [17, 18]. Additionally, lactic acid bacteria isolated from honey samples including *Lactobacillus plantarum*, *Lactobacillus curvatus*, *Pediococcus acidilactici*, and *Pediococcus pentosaceus* showed inhibition against pathogenic *Candida* species [19]. Both lactic acid bacteria and *Bacillus* spp. produce a variety of antifungal secondary metabolites including organic acids, volatile compounds, ribosomally synthesized peptides, and nonribosomal peptides [20–22]. The potential application of these microbial natural products in the food industry, agricultural and medical field is promising. One example is nisin, a bacteriocin isolated from *Lactococcus lactis* subsp. *lactis* strain and exhibits broad-spectrum antibacterial activity [21]. Nisin is used in dairy and meat products as

a biopreservative compound to inhibit foodborne pathogen *Listeria monocytogenes* [23]. Additionally, several strains of *B. subtilis*, *B. thuringiensis*, and *B. amyloliquefaciens* were approved as commercial biopesticides by the Environmental Protection Agency (EPA) [24]. Lipopeptides secreted by these *Bacillus* species were used commercially as antifungal agents to control plant diseases caused by phytopathogens [25].

In an effort to isolate novel antifungal compounds as candidates for medical and/or agricultural applications, we designed this study to isolate, purify, and characterize antifungal proteinaceous compounds from raw honey. Several *Bacillus* strains were isolated from raw clover and orange blossom honey. Extracellular antifungal compounds were purified via ammonium sulfate precipitation, solid phase extraction (SPE), and reversed-phase high performance liquid chromatography (RP-HPLC). Whole genome sequencing was performed on two antifungal producing strains identified as *B. velezensis*. Using a combination of genome secondary metabolite gene cluster analysis and mass spectrometry (MS), we determined that the antifungal compound belonged to the iturin family.

## Results

Four of 15 bacterial isolates from clover honey and 8 of 23 isolates from orange blossom honey yielded an inhibition zone when spotted on at least one fungal indicator. The 16S rRNA gene sequence of these 12 isolated strains showed highest identity to that of several *Bacillus* spp. To evaluate the antifungal potential of honey isolates, food-isolated fungal strains were selected as indicators for antifungal assay (S1 Table). Cross reactivity of the honey bacterial isolates against these fungal strains and BLAST identification results were summarized in Table 1.

**Table 1. Summary of identity, source, and cross-reactivity against food-associated fungal indicators of honey bacterial isolates.**

| Isolates | BLAST ID [a] | Honey Source | Cross reactivity[b] | | | | | | | |
|---|---|---|---|---|---|---|---|---|---|---|
| | | | *Syncephalastrum* | *Aspergillus* S11-0016 | *Aspergillus* S11-0033 | *A. fumigatus* | *A. niger* | *Rhodotorula* | *P. glabrum* | *Cladosporium* |
| Co-1 | *B. toyonensis* | Clover | - | - | - | + | - | - | - | - |
| Co-5 | *B. toyonensis* | Clover | - | - | + | ++ | ++ | - | ++ | - |
| Co-6 | *B. toyonensis* | Clover | + | - | + | ++ | ++ | + | ++ | - |
| Co-10 | *B. aerius* | Clover | \ | \ | \ | + | - | \ | + | + |
| Co-17 | *B. cereus* | Orange blossom | + | - | + | ++ | + | - | + | - |
| Co-18 | *B. megaterium* | Orange blossom | \ | \ | \ | - | - | \ | - | - |
| Co-20 | *B. amyloliquefaciens* | Orange blossom | + | \ | - | + | + | + | - | ++ |
| Co-21 | *B. cereus* | Orange blossom | + | - | + | + | - | - | - | - |
| Co-26 | *B. amyloliquefaciens* | Orange blossom | + | \ | \ | + | - | - | + | + |
| Co-29 | *B. amyloliquefaciens* | Orange blossom | + | \ | \ | + | + | - | - | ++ |
| Co-30 | *B. amyloliquefaciens* | Orange blossom | + | \ | \ | + | + | - | - | ++ |
| Co-33 | *B. aryabhattai* | Orange blossom | \ | \ | \ | - | - | \ | - | - |

[a] BLAST ID was determined based on 16S rRNA gene homology search using NCBI Nucleotide BLAST tools. The species with the highest similarity were reported.

[b] Cross reactivity was determined using deferred overlay inhibition assay. The inhibition level against the fungal indicators was defined based on visual observation. "+": low inhibition level. "++": strong inhibition level. "-": no observed inhibition. "\": inconclusive result.

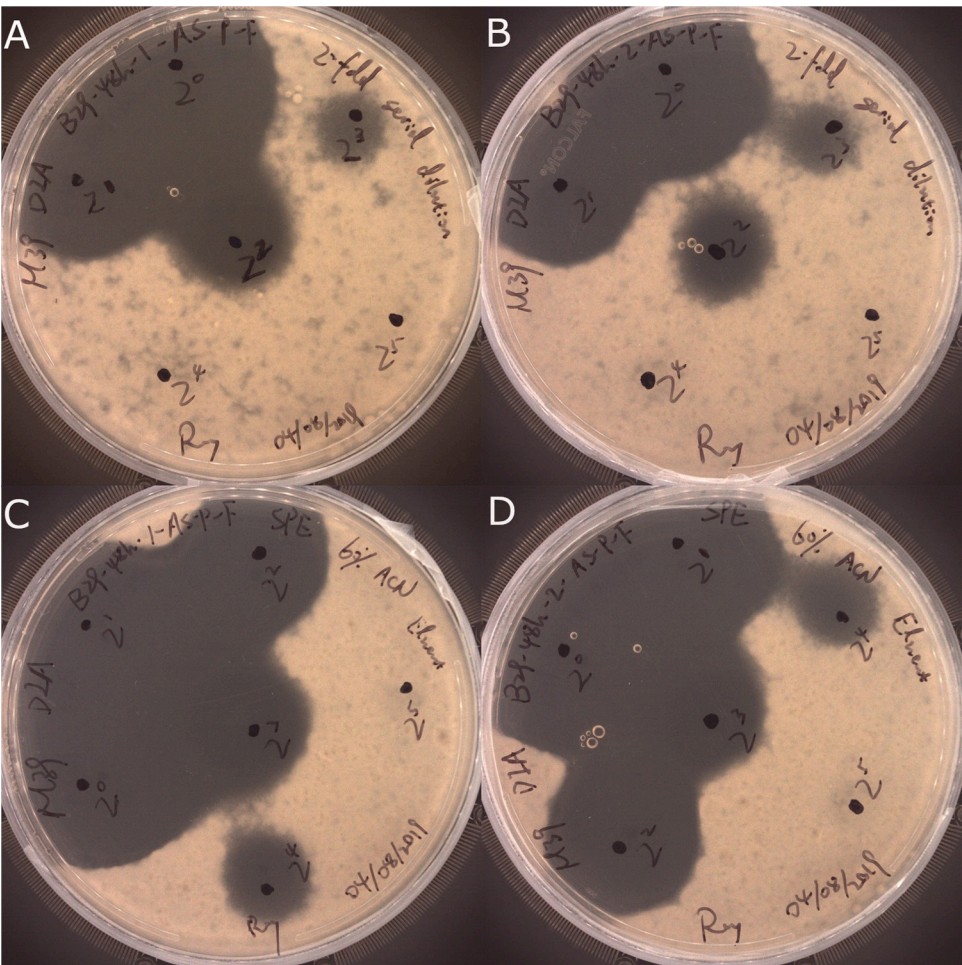

**Fig 1. Deferred inhibition assay of purified products from *Bacillus velezensis* WRB-ZX-001 and WRB-ZX-002 against food-isolated *Aspergillus fumigatus*.** Precipitates of WRB-ZX-001 and WRB-ZX-002 from 60% ammonium sulfate were shown in A and B. Solid phase extraction eluates of 60% acetonitrile for WRB-ZX-001 and WRB-ZX-002 were shown in C and D. Two-fold serial dilution was performed for all samples to determine the antifungal activity units.

Isolates that showed antifungal activity against at least three fungal indicators were selected for antifungal production in liquid broth. The production conditions, including the medium type, incubation temperature, and shaking speed, were optimized. As the only two isolates showing the ability to excrete antifungal compounds, isolate Co-29 and Co-30 were selected and renamed as WRB-ZX-001 and WRB-ZX-002 for the following experiments. These two isolates were grown in BHI broth at 30°C, 150 rpm for 24 hours and 48 hours, and cell-free supernatant showed clear inhibition zones against fungal indicators. The antifungal compounds produced by the isolates were further purified and the isolates were whole genome sequenced.

Antifungal compounds produced by isolates WRB-ZX-001 and WRB-ZX-002 were first purified by ammonium sulfate precipitation of the cell-free culture supernatant. Precipitate from 60% ammonium sulfate showed the highest antifungal activity (Fig 1, Table 2). Ammonium sulfate precipitates were further purified by solid phase extraction with C18 columns and acetonitrile. The eluants for the optimal recovery of antifungal compounds were 50% and 60% acetonitrile. The SPE eluates were loaded onto HPLC, and fractions were collected to test for antifungal activity. Two major peaks were observed in the HPLC spectra and fractions with

**Table 2. Antifungal activity of purification products of *Bacillus velezensis* isolates against food-isolated *Aspergillus fumigatus*.**

| Purification procedure | *Bacillus velezensis* WRB-ZX-001 | *Bacillus velezensis* WRB-ZX-002 |
|---|---|---|
| Cell-free filtrate | 20 AU/mL | 40 AU/mL |
| Ammonium sulfate precipitant | 800 AU/mL | 800 AU/mL |
| Solid phase extraction eluate | 800 AU/mL | 1600 AU/mL |
| HPLC fraction | 200 AU/mL | 200 AU/mL |

Antifungal activity unit (AU/mL) is defined as the reciprocal of the highest dilution showing a clear inhibition zone.

elution times between 28.5 min to 30 min for both isolates showed highest antifungal activity (200 AU/mL). These fractions were loaded once more onto HPLC to confirm their purity, and single peak was observed for both samples (Fig 2). SPE eluates (50% acetonitrile) and HPLC fraction collection samples (28.5 min to 30 min) for isolates WRB-ZX-001 and WRB-ZX-002 were analyzed with DIMS and LC-MS, respectively. The HPLC samples analyzed with LC-MS showed major peaks with *m/z* value of 1057.57 (Fig 3), which was also present in SPE sample WRB-ZX-002 (results not shown). Another compound with singly charged *m/z* value of 1043.55 and doubly charged *m/z* value of 522.28 was present in both SPE eluates and HPLC collected samples (S1 Fig). Based on results from previous studies, we presumed that the ions with *m/z* value of 1043.55 and 1057.57 were $C_{14}$ and $C_{15}$ iturin A $[M+H]^+$, respectively [26, 27]. The molecular formula of $C_{14}$ and $C_{15}$ iturin A is $C_{48}H_{74}N_{12}O_{14}$ and $C_{49}H_{76}N_{12}O_{14}$ [28].

For isolate WRB-ZX-001, the 4,183,488 bp genome was assembled to 15 contigs with an average coverage of 104x and N50 of 685,546 bp. For isolate WRB-ZX-002, the genome size is 4,185,188 bp, and the genome was assembled to 15 contigs with an average coverage of 128x and N50 of 1,001,971 bp. Both isolates have the same GC content of 45.97%. Isolate WRB-ZX-001 contains an estimated 4,165 genes and 4,003 coding sequences (CDSs), while isolate WRB-ZX-002 contains an estimated 4,167 genes and 4,004 CDSs. To obtain functional labels, protein BLAST hits were mapped against the curated Gene Ontology (GO) database and GO

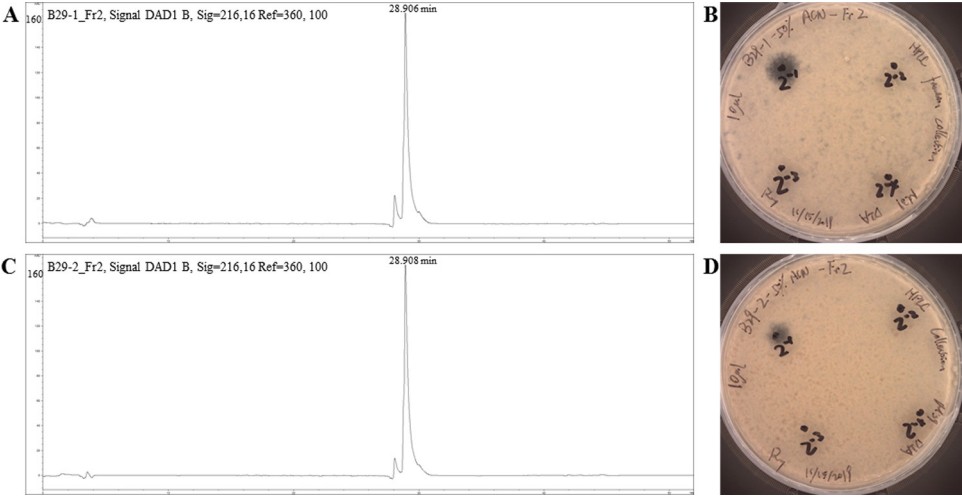

**Fig 2. Reversed-phase HPLC of purified products of *Bacillus velezensis* WRB-ZX-001 and WRB-ZX-002.**
Purification process included ammonium sulfate precipitation, solid phase extraction, and HPLC fraction collection. Single peaks shown in A and C were from isolate WRB-ZX-001 and WRB-ZX-002, respectively, and both have shown inhibition against fungal indicator strain *Aspergillus fumigatus* as shown in B and D.

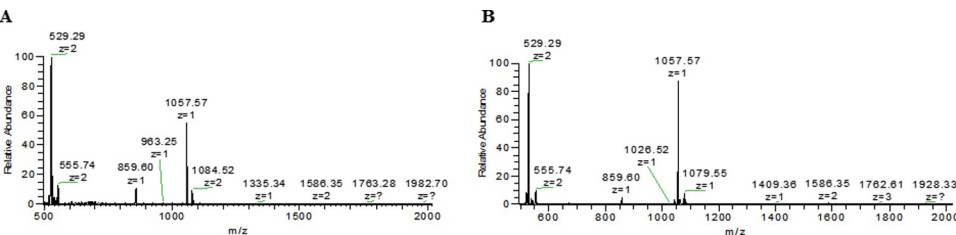

**Fig 3. Mass spectra for purified antifungal compounds produced by *Bacillus velezensis* WRB-ZX-001 and WRB-ZX-002.** A and B are LC-MS spectra for HPLC collected active fraction of WRB-ZX-001 and WRB-ZX-002. Ion with *m/z* value of 1057.57 was assigned to $C_{15}$ iturin A $[M+H]^+$. Ion with *m/z* value of 1079.55 was assigned to $C_{15}$ iturin A $[M+Na]^+$.

terms were assigned to the query sequences, with 3,261 annotated sequences for isolate WRB-ZX-001 and 3,263 annotated sequences for isolate WRB-ZX-002. Based on BLAST2GO genome annotation results, the predicted CDSs were assigned to three principal categories: biological process, cellular component, and molecular function. For isolates WRB-ZX-001 and WRB-ZX-002, the most abundant groups in the category of biological process were cellular process (38%), metabolic process (36%), and biological regulation (8%). In the category of cellular component, the most dominant terms were integral component of membrane (55%), cytoplasm (26%), and plasma membrane (16%). In the category of molecular function, the most representative terms were hydrolase activity (32%), oxidoreductase activity (17%), metal ion binding (13%), transmembrane transporter activity (11%), DNA binding (10%), and ATP binding (9%). Detailed GO annotation and node score distribution for *Bacillus velezensis* WRB-ZX-001 and WRB-ZX-002 was reported in S3 Table. To calculate average nucleotide identity (ANI) and classify the two isolates at species level, orthoANI analysis was performed. The type strain that isolates WRB-ZX-001 and WRB-ZX-002 were most closely related to was *B. velezensis* FZB42, with orthoANI values of 98.96% and 98.93% respectively. Based on the proposed species boundary of 95–96% orthoANI value, we concluded that both WRB-ZX-001 and WRB-ZX-002 should be classified as *B. velezensis* species [29–31]. To elucidate the phylogenetic relationships between our two isolates and the closely related *B. amyloliquefaciens* group, a total of 42 reference genomes were obtained from the NCBI database. Forty-one *B. amyloliquefaciens* group isolates and one *B. subtilis* subsp. *subtilis* str. 168 were included in the phylogenetic analysis. The phylogenetic tree based on 4,035 core genome SNPs revealed close relatedness of the two isolates from this study with type strains *B. velezensis* FZB42 and *B. velezensis* KACC18228 (Fig 4). Additional genome comparison of *B. velezensis* type strains FZB42 and CBMB205, *B. amyloliquefaciens* type strain DSM7, and isolates WRB-ZX-001 and WRB-ZX-002 was visualized with BRIG version 0.95 (Fig 5). Gaps in the circular chromosome represented regions with no homology to the reference strain *B. velezensis* FZB42. Gaps for the two isolates from our study were consistent due to high levels of nucleotide homology. Several gaps were present when comparing two isolates from this study with the closely related type strain *B. velezensis* FZB42, indicating the potential presence of novel gene products. To evaluate the secondary metabolite synthesis potential, genomes of WRB-ZX-001 and WRB-ZX-002 were annotated using NCBI Prokaryotic Genome Annotation Pipeline (PGAP) database. BAGEL4 was used to predict open reading frames (ORFs) for ribosomally synthesized proteins and peptides, including bacteriocins, ribosomally synthesized and post-translationally modified peptides (RiPPs). Five putative gene clusters of interest were identified by BAGEL4 in the genomes of *B. velezensis* WRB-ZX-001 and WRB-ZX-002. Both strains contained 3 contigs with genes related to the production of secondary metabolites, including antimicrobial peptide

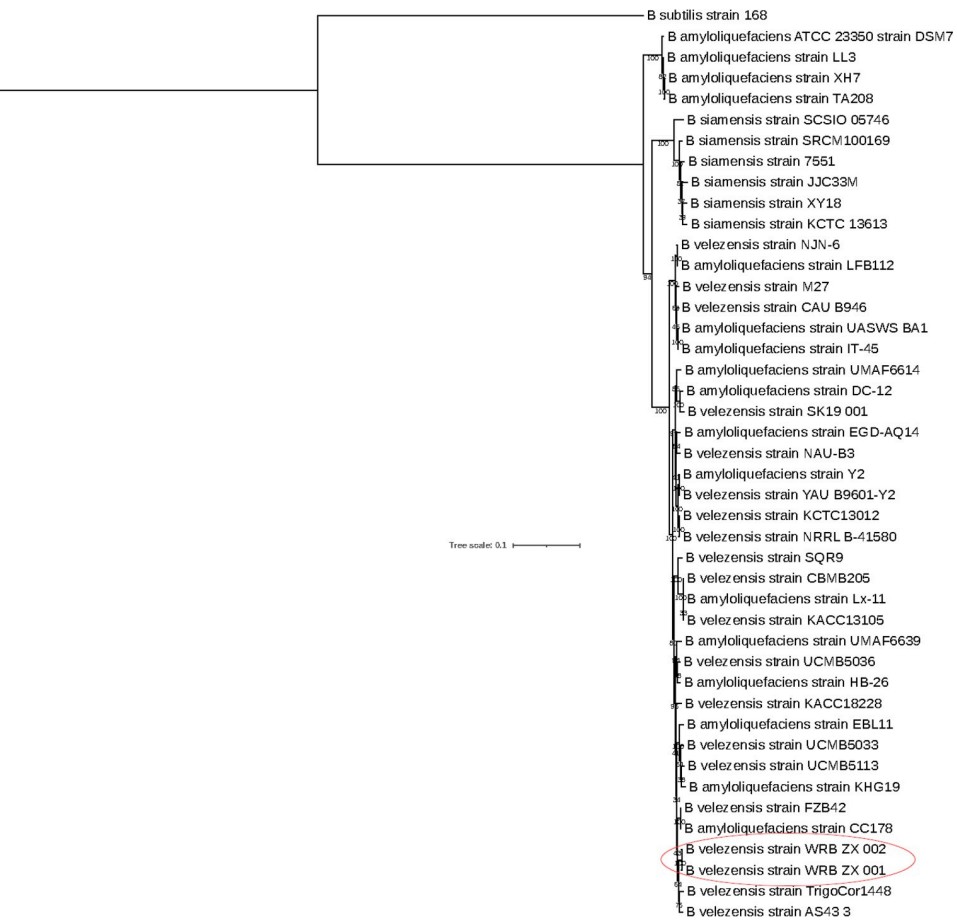

**Fig 4. Core genome phylogeny of 43 *Bacillus amyloliquefaciens* group isolates.** Maximum likelihood tree was constructed with core genome SNPs identified by kSNP. 41 reference genomes of *Bacillus amyloliquefaciens* group isolates were obtained from NCBI genome database. The core genome of *Bacillus subtilis* 168 was used as outgroup. Phylogeny was inferred by RAxML under time-reversible model with gamma distributed substitution sites and 1000 bootstrap repetitions. Bar represents 0.2 substitution per site. Isolates from this study were labeled with red circle.

LCI and thiopeptide, bacteriocin amylocyclicin, linear azole/azoline-containing peptide (LAP), and lantibiotic cerecidin. Additionally, antiSMASH was used to identify secondary metabolite biosynthetic gene clusters (BGCs) including nonribosomal peptide synthetases (NRPSs), polyketide synthases (PKSs), RiPPs, and other antimicrobial synthases. A total of 16 putative BGCs were identified in both genomes, including 5 NRPSs for bacillibactin, fengycin, bacillomycin D, iturin and surfactin, three trans-acyl-transferase polyketide synthases (trans-AT-PKS) for macrolactin H, bacillaene and difficidin, one type III PKS, three RiPP clusters for thiopeptide, lanthipeptide, amylocyclin, and others (Table 3). According to the results of anti-SMASH analysis, both isolates contained a gene cluster with 88% similarity to iturin synthetase, and the predicted peptide sequence of the nonribosomal peptide is L-Asn-D-Tyr-D-Asn-L-Gln-L-Pro-D-Asn-L-Ser. To further confirm the presence of iturin gene cluster, BLAST analysis was performed on both genomes. Four iturin genes (*ituD*, *ituA*, *ituB*, *ituC*) were detected in the genome of both isolates, with a similarity of 98.60% to *itu* operon complete CDS from the reference strains *B. subtilis* ZK0 (NCBI accession number: KT781920.1) and *B. subtilis* subsp. *krictiensis* str. ATCC 55079 (NCBI accession number: KU170613.1). The presence of iturin gene clusters in the genome further validated the MS data, indicating the production of $C_{14}$-iturin (*m/z* of $[M+H]^+$ 1043.55) and $C_{15}$-iturin (*m/z* of $[M+H]^+$ 1057.57).

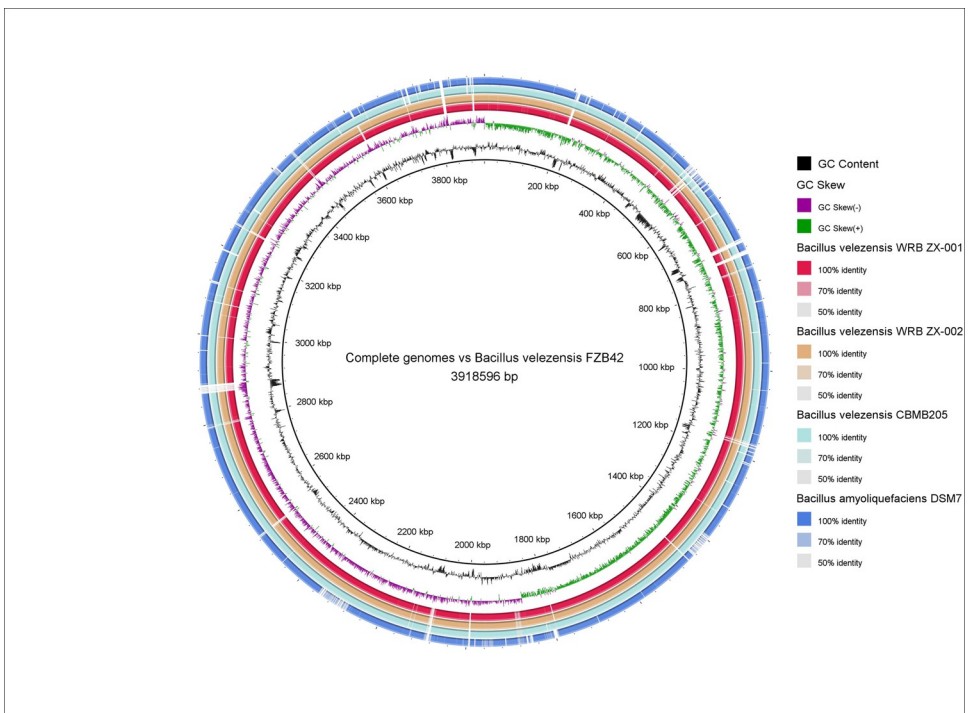

**Fig 5. Genome comparison of *Bacillus velezensis* WRB-ZX-001 and WRB-ZX-002 against closely related *Bacillus* type strains.** *Bacillus velezensis* FZB42 was used as the reference strain. The circular ring map was constructed by BLAST Ring Image Generator (BRIG, version 0.95). From inner to outer ring: 1) GC content; 2) *Bacillus velezensis* FZB42 nucleotide sequence; 3) GC Skew; 4) *Bacillus velezensis* WRB-ZX-001 nucleotide sequence; 5) *Bacillus velezensis* WRB-ZX-002 nucleotide sequence; 6) *Bacillus velezensis* CBMB205 nucleotide sequence; 7) *Bacillus amyloliquefaciens* DSM7 nucleotide sequence.

Absorption at OD 600 nm was used to plot the growth curve for isolate WRB-ZX-001 and WRB-ZX-002. Antifungal activity against fungal indicator *A*. *fumigatus* was calculated and plotted with the growth curve (Fig 6). The antifungal production started at 24 hours and 18 hours for WRB-ZX-001 and WRB-ZX-002, respectively. The highest antifungal production occurred after cells reached late stationary phase. The antifungal activity was quantified by serial dilution, which was the reason why the antifungal activity fluctuated before reaching maximum production. To optimize the production of antifungal compounds, bacterial cells were collected at 48 hours for the following experiments. The heat stability and protease stability for the antifungal compounds produced by WRB-ZX-001 and WRB-ZX-002 were tested and results were summarized in Table 4. After heat treatment using a 15 min, 121˚C cycle in the autoclave, a 2-fold decrease in antifungal activity for WRB-ZX-001 was observed while sample WRB-ZX-002 had no decrease. For the protease stability test, antifungal compounds produced by WRB-ZX-001 and WRB-ZX-002 showed resistance to pronase E, α-chymotrypsin, and trypsin, with no change in their antifungal activity compared to control. Only sample WRB-ZX-002 showed a 2-fold decrease in antifungal activity after treatment with pepsin. Based on these results, we concluded that antifungal compounds produced by WRB-ZX-001 and WRB-ZX-002 were heat-resistant and protease-resistant.

## Discussion

In general, bacterial spores are abundant in raw honey, many of which have the potential to exhibit antifungal properties [19]. Previous studies have isolated *Bacillus* spp., *Clostridium*

**Table 3. Potential secondary metabolite synthesis gene clusters identified in *Bacillus velezensis* WRB-ZX-001 and WRB-ZX-002 by antiSMASH 5.0.**

| Strain | Cluster | Type | From[a] | To[a] | Secondary metabolite | Similarity[b] (%) |
|---|---|---|---|---|---|---|
| *Bacillus velezensis* WRB-ZX-001 | 1 | Other | 298857 | 354273 | Bacilysin | 100 |
| | 1 | Other | 500231 | 554177 | Teichuronic acid | 100 |
| | 1 | NRPS | 876498 | 928287 | Bacillibactin | 100 |
| | 1 | RiPP | 876498 | 928287 | Amylocyclicin | 100 |
| | 2 | PKS-like | 65404 | 106648 | \ | \ |
| | 2 | Terpene | 189448 | 210188 | \ | \ |
| | 2 | TransAT-PKS | 557837 | 646070 | Macrolactin H | 100 |
| | 3 | T3PKS | 212154 | 250873 | \ | \ |
| | 3 | Terpene | 314561 | 336444 | \ | \ |
| | 3 | NRPS | 360563 | 498152 | Fengycin/Plipastatin | 100 |
| | 3 | NRPS | 360563 | 498152 | Bacillomycin D | 100 |
| | 3 | NRPS | 360563 | 498152 | Iturin | 88 |
| | 3 | TransAT-PKS | 560507 | 670621 | Bacillaene | 100 |
| | 4 | TransAT-PKS | 85797 | 191987 | Difficidin | 100 |
| | 5 | Thiopeptide/LAP | 114108 | 143862 | \ | \ |
| | 5 | NRPS | 154585 | 219992 | Surfactin | 91 |
| | 8 | Class II lanthipeptide | 47789 | 66288 | \ | \ |
| *Bacillus velezensis* WRB-ZX-002 | 1 | TransAT-PKS | 550815 | 656586 | Difficidin | 100 |
| | 1 | T3PKS | 954998 | 993717 | \ | \ |
| | 1 | Terpene | 1057405 | 1079288 | \ | \ |
| | 1 | NRPS | 1103407 | 1240996 | Fengycin/Plipastatin | 100 |
| | 1 | NRPS | 1103407 | 1240996 | Bacillomycin D | 100 |
| | 1 | NRPS | 1103407 | 1240996 | Iturin | 88 |
| | 1 | TransAT-PKS | 1303351 | 1413465 | Bacillaene | 100 |
| | 2 | NRPS | 73679 | 125468 | Bacillibactin | 100 |
| | 2 | Other | 447789 | 501741 | Teichuronic acid | 100 |
| | 2 | Other | 647699 | 703115 | Bacilysin | 100 |
| | 3 | PKS-like | 65404 | 106648 | \ | \ |
| | 3 | Terpene | 189448 | 210188 | \ | \ |
| | 3 | TransAT-PKS | 557837 | 646070 | Macrolactin H | 100 |
| | 4 | Thiopeptide/LAP | 114001 | 143734 | \ | \ |
| | 4 | NRPS | 154457 | 219864 | Surfactin | 91 |
| | 6 | Class II lanthipeptide | 1 | 18500 | \ | \ |

[a] Location of gene clusters in the *Bacillus velezensis* genome.

[b] Similarity based on BLAST analysis against known gene clusters.

spp., *Lactobacillus* spp. and other lactic acid bacteria (LAB) from raw honey [32]. Many members from LAB and *Bacillus* species have been shown to be antifungal, including *Lactobacillus casei*, *Lactobacillus plantarum*, *B. subtilis* and *B. velezensis* [22]. Bioactive compounds, like ribosomally synthesized bacteriocins and non-ribosomally synthesized small peptides, can be produced by these bacteria, which could potentially be exploited for industrial and medical applications. In this study, our two antifungal *B. velezensis* isolates from raw honey are inhibitory against various food-isolated fungi (Table 1). *Bacillus* species devote a large portion of their genome to secondary metabolism, potentially due to competition they face in the environment [33]. *Bacillus* species are ubiquitous in soil and the ocean, which often have complex microbial communities. By producing secondary metabolites that can inhibit closely related

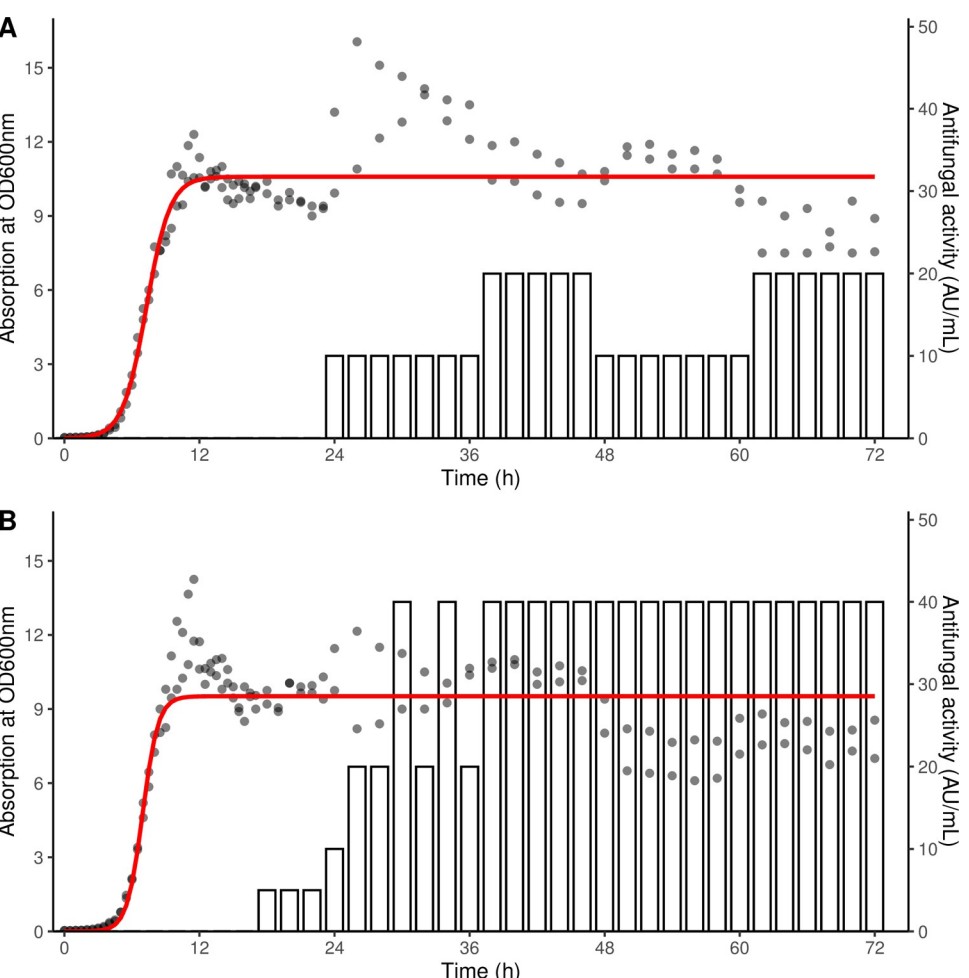

**Fig 6.** Growth curve and antifungal activity curve for *Bacillus velezensis* WRB-ZX-001 (A) and WRB-ZX-002 (B). Growth curve was plotted by measuring absorption at $OD_{600nm}$ every 30 min and a standard form of logistic equation was used to fit the absorption data (red line). Antifungal activity was measured by well diffusion overlay inhibition assay of serially diluted cell-free supernatant every two hours against fungal indicator strain *Aspergillus fumigatus* and data were shown in bar plots.

species and other microorganisms in the ecological niche, *Bacillus* species have gained significant survival advantages [34]. Previous researchers have isolated a variety of secondary metabolites with antibacterial and antifungal properties from *Bacillus* species, some of which are nonribosomal peptides (NRPs) [34]. NRPs are synthesized by nonribosomal peptide

**Table 4. Antifungal activity of heat-treated and protease-treated purified products of *Bacillus velezensis* isolates against food-isolated *Aspergillus fumigatus*.**

| Treatment | *Bacillus velezensis* WRB-ZX-001 | *Bacillus velezensis* WRB-ZX-002 |
|---|---|---|
| Control | 800 AU/mL | 800 AU/mL |
| 121˚C, 15 min | 400 AU/mL | 800 AU/mL |
| Pronase E | 800 AU/mL | 800 AU/mL |
| Chymotrypsin | 800 AU/mL | 800 AU/mL |
| Pepsin | 800 AU/mL | 400 AU/mL |
| Trypsin | 800 AU/mL | 800 AU/mL |

synthetases (NRPSs) and independent of messenger RNA. NRPs usually go through extensive modifications, including glycosylation, acylation, and hydroxylation. Due to these modifications, some NRPs are amphiphilic and able to insert into cell membrane to form pores, like gramicidin, surfactin, fengycin, iturin, and other lipopeptides. Pore formation in cell membrane will lead to ion leakage and cell death [35, 36]. Some NRPs target closely related cells while others have broad spectrum. Taking account of the results from LC-MS (Fig 3) and secondary metabolite genome mining pipeline (BAGEL4 and antiSMASH) (Table 3), we determined that the major broad-spectrum antifungal compound produced by our *B. velezensis* isolates was a nonribosomal lipopeptide, iturin A.

The iturin A operon was demonstrated to contain four open reading frames (ORFs): *itu*D, *itu*A, *itu*B, and *itu*C. *itu*D encodes a putative malomyl coenzyme A transacylase, while *itu*A, *itu*B, and *itu*C encode iturin synthetases [37]. Iturin A production is regulated by the promoter on the upstream of *itu*D [37]. All four ORFs as well as the promoter $P_{itu}$ were present in the genome of our two *B. velezensis* isolates based on BLAST search, with an identity of 98.6% to *itu* operon complete CDS. The iturin family is a group of cyclic lipopeptides with hydrophilic C-terminal heptapeptides and characteristic hydrophobic N-terminal β-amino fatty acids. The aliphatic chain of iturin contains between 14 to 17 carbons and the peptide chain has a chiral sequence of LDDLLDL [38]. The iturin family primarily has broad-spectrum antifungal activity, with limited antibacterial activity [39]. In our study, iturin-producing *B. velezensis* strains showed broad spectrum antifungal activities, with antagonistic ability against *Aspergillus*, *Cladosporium*, *Syncephalastrum* (Table 1), and *Candida albicans* (results not shown). The proposed antifungal mechanism for the iturin family is that they can interact with sterol components on the surface of fungal membrane and increase potassium permeability [40]. Previous studies showed that iturin A can form ion-conducting pores on bimolecular lipid membranes and cholesterol can facilitate the pore-formation by expanding the open-state lifespan [35, 36, 41]. Additionally, iturin is able to self-associate and interact with lipid membranes by forming a stoichiometric complex with cholesterol on the membrane surface [40]. Furthermore, iturins with longer acyl chains have stronger antifungal properties due to their ability to form oligomers and insert deeply into target membranes to form ion-conducting pores [42]. The pore-forming and membrane permeabilizing abilities of iturin A is concentration dependent. At high concentrations, iturin A showed higher antagonistic activity against fungal cells and higher hemolytic activity [43].

In previous studies, *Bacillus* species have been demonstrated to be able to produce antifungal lipopeptides including members from iturin family. In a study by Pathak and Keharia (2014), iturin isomers and surfactin families were isolated from crude extract of *B. subtilis*. Iturin A2 and Iturin A3/A4/A5 were found to have broad spectrum antifungal activities against *Aspergillus*, *Fusarium*, *Chrysosporium*, *Candida albicans*, *Trichosporium*, *Alternaria*, and *Cladosporium* [26]. One of the iturin A homologues in their study had a mass of 1057.5, the same as the iturin isolated from our study. Similar to the results from our research, Gong et al. (2006) identified antifungal lipopeptides from *B. subtilis* strain PY-1 that was temperature stable and protease resistant. By using ESI-TOF MS, FAB-MS/MS CID spectrometry and NMR, they identified the antifungal compounds as iturin A isomers and determined that the (M+H)$^+$ ions at *m/z* 1057 were iturin A3 and A4 (C17 aliphatic chain) [44]. Moreover, another group of researchers isolated *B. amyloliquefaciens* S76-3 from wheat spikes, which produced antifungal lipopeptides active against *Fusarium graminearum*. These lipopeptides were identified through RP-HPLC and ESI-MS, with iturin A and plipastatin A being the most abundant molecules. The *m/z* value of iturin A with C-14 acyl acid chain was 1043.35. Fluorescence microscopy analyses and transmission electron microscopy (TEM) analyses of lipopeptide-treated *Fusarium graminearum* conidia and hyphae showed damages to cell wall and plasma

membrane, which was consistent with the proposed antifungal mechanism of iturin family [45]. Overall, *itu operon* is common in *B. subtilis* group and *B. amyloliquefaciens* group, and our *B. velezensis* isolates were demonstrated to possess *itu* operon and produce $C_{14-15}$ iturin A.

The production of iturin and other lipopeptides by *Bacillus* species is dependent on the environment factors, including temperature, pH, carbon source, and oxygen availability. Iturin is mainly produced at temperature between 25˚C and 37˚C under aerobic conditions [46]. In our study, the optimum temperature for the production of iturin A by *B. velezensis* strains was 30˚C. A neutral pH is generally favorable for the production of lipopeptides [43]. In our study, iturin production was optimized by adjusting the pH of BHI broth to 7.4. In a recent study by Dang et al (2019), the optimal condition for the iturin A production by *B. amyloliquefaciens* LL3 derivative strain was thoroughly investigated using single factor optimization and response surface methodology. It was determined that inulin was the best carbon source and L-sodium glutamate was the best nitrogen source. The optimal production condition was determined to be pH 7.0 and 27˚C with 7, 15 and 0.5 g/L of inulin, L-sodium glutamate and $MgSO_4$ [47]. In our future studies, this condition will be validated to optimize the production of iturin by our *B. velezensis* isolates.

Compared to conventional synthetic fungicides, which raise concerns regarding chemical residues and antibiotic resistance, biocontrol agents synthesized by living organisms are relatively more environmentally friendly for agricultural applications [48, 49]. Lipopeptides, like iturin, are considered safe, biodegradable, and eco-friendly. Some previous studies have demonstrated their potential application. Lipopeptides produced by *B. subtilis* RB14, containing iturin A and surfactin, were effective at suppressing the damping-off of tomato seedings cause by *Rhizoctonia solani*. A mutant of *B. subtilis* RB14 that cannot produce iturin A or surfactin failed to inhibit *R. solani*. Restoration of the gene successfully reinstated the suppressibility toward the fungal disease [50]. Another study constructed a mutant of *B. subtilis* ATCC6633 by replacing native promoter with constitutive promoter to increase the production of mycosubtilin. The mutant strain was able to reduce *Pythium* infection in tomato seedlings and increase germination rate [51]. Romero et al (2007) showed in their study that direct application of lipopeptide-producing *B. subtilis* cells or cell-free filtrate to leaf surface can prevent powdery mildew caused by *Podosphaera fusca*. Furthermore, by using site-directed mutagenesis, they demonstrated that bacterial mutants that lost the ability to produce bacillomycin, fengycin or iturin A were not able to control the powdery mildew disease [52]. Antifungal lipopeptides produced by *Bacillus* species could have additional applications. These lipopeptides possess the ability to change biofilm formation, motility, and virulence gene expression of various microorganisms. It is also associated with plant root colonization, plant defense, and plant growth promotion [53]. With the increased need for biopesticides that have high specificity, low environmental persistence, and low toxicity, industrial exploitation of these chemicals or compounds derived from natural products as food preservatives and crop protection agents is continuously expanding [54]. Iturin A, a biopesticide produced by food-isolated *Bacillus* spp. and naturally present in food systems, can be exploited for industrial applications.

The safety of the producer strains and their products need to be evaluated to achieve broader application of iturin A produced by *Bacillus* species. One of the major producer strains for iturin A, *B. subtilis*, has Qualified Presumption of Safety (QPS) status according to European Food Standards Authority (EFSA), which indicates that this strain does not harbor acquired antimicrobial resistance (AMR) genes or exhibit toxigenic activity [34]. *B. velezensis* FZB42 type strain, which is closely related to our isolates, was also evaluated by US Environmental Protection Agency (EPA) and considered not toxic, pathogenic, or infective. Therefore, a tolerance exemption for residues of *B. velezensis* FZB42 in food commodities was established. As for the surface-active agents produced by these strains, including surfactin, iturin and other

detergents, they can penetrate cell membrane but are not necessarily cytotoxic. Toxicity assays need to be developed to determine their cytotoxicity specifically. For now, current safety measures taken by the industry, including historical safety data and routine testing of the strains and products, are sufficient to ensure the safe usage of strains from the *B. subtilis* and *B. amyloliquefaciens* group as enzyme production workhorse [34]. Regarding the safety of iturin A, the acute and subacute toxicity was previously evaluated in mouse models. Preliminary toxicology study showed that iturin A can induce hepatotoxicity and was deposited in liver, lung, and spleen. However, organ-specific toxicity of iturin A was reversible after discontinuation of treatment, which indicated that medical application is still possible [55]. On the other hand, in a study by Zhao et al (2018), iturin produced by *B. subtilis* was intragastrically administered to mouse models. In acute (7-day) and subacute (28-day) toxicity tests under concentration of 5000 mg/kg and 2000 mg/kg, respectively, iturin was deemed safe and non-toxic, with no significant damage to liver, kidney, or small intestines [56]. Overall, rigorous and large scale *in vivo* and clinical studies are still needed to fully understand the potential toxicity of iturin A.

A large portion of recent publications on antifungal lipopeptides produced by *Bacillus* species and other gram-positive bacteria focused on partially purified mixtures with varying antifungal activities. The chemical identities of these semi-purified compounds remain uncharacterized, and the biological implications of these studies remain unclear, which poses barriers to future studies. Moreover, with the increased availability and popularity of next-generation sequencing (NGS) and genome mining tools, more recent studies are using these tools to evaluate the secondary metabolites produced by *Bacillus* species. However, chemical confirmation of those potential metabolites is falling behind. Future studies need to combine genetic and genomic methods with traditional chemical identification methods to properly identify, classify, and characterize these potential secondary metabolites. As for the future of iturin family, additional studies are necessary to improve the production efficiency and evaluate the resistance development in fungal model systems.

## Materials and methods

### Antifungal isolates selection

Raw clover honey and orange blossom honey were purchased from a local honey shop (Dundee, NY). Honey samples were diluted with 0.1% peptone water, and 100 μL of $10^{-1}$ and $10^{-2}$ dilutions were spread plated on tryptic soy agar (TSA) (BD Difco, Franklin Lakes, NJ). Plates were incubated at 30°C for 24 hours. Visually distinct colonies were selected to test their antifungal activity. Eight fungal strains isolated from commercially processed food products were used as antifungal activity indicators [57]. Food-isolated fungal strains were incubated at ambient temperature on potato dextrose agar (PDA, BD Difco, Franklin Lakes, NJ) for at least 4 weeks prior to harvest. Fungal spores were harvested by flooding the surface of fully grown plates with 10 mL 0.1% Tween 80 (Sigma, St. Lois, MO). Spore suspension was filtered with several layers of sterile cheese cloth to remove debris and stored at -80°C.

### Antifungal assays

Antifungal activities of bacterial isolates were determined by deferred overlay inhibition assay: fungal spore suspensions were mixed with 10 mL 0.75% soft TSA and overlaid on PDA plates. Bacterial isolates were spotted with sterile toothpicks on the surface of solidified soft agar with fungal indicators. Plates were incubated at ambient temperature for 48 to 72 hours and inhibition zones were recorded. Bacterial colonies that showed antifungal properties were selected for further analysis. Bacterial isolates were stored in 20% glycerol at -80°C.

## Bacterial classification through 16S rRNA gene sequencing

Bacterial isolates exhibiting strong inhibition toward fungal indicators were initially identified by 16S rRNA gene sequencing. DNA was obtained using the Genomic DNA extraction kit (Qiagen, Germantown, MD) and 16S rRNA genes were amplified through polymerase chain reaction (PCR). A set of primers (IDT, Coralville, IA) were used to amplify the conserved region in bacteria. 16S forward primer sequence: 5'-AGAGTTTGATCCTGGCTCAG-3'. 16S reverse primer sequence: 5'- AAGGAGGTGATCCAGCC-3'. PCR procedures were as follows: 3 μL DNA template, 1 μL forward primer and 1 μL reverse primer, 0.3 μL GoTaq Flexi DNA polymerase (Promega, Madison, WI), 10 μL 5X Colorless GoTaq Flexi buffer (Promega, Madison, WI), 4 μL 25 mM $MgCl_2$ (Promega, Madison, WI), 2 μL 10 mM dNTP (New England Biolabs, Ipswich, MA), 29 μL $dH_2O$. Total volume was 50 μL per PCR tube. Thermal cycling conditions were as follows: 1 cycle of 94˚C for 5 minutes, 35 cycles of 94˚C for 30 seconds, 50˚C for 1 minute, 72˚C for 2 minutes, 1 cycle of 72˚C for 10 minutes. PCR products were purified by QIAquick PCR purification kit (Qiagen, Germantown, MD). Purified DNA products were sent to Cornell University Biotechnology Resource Center (Ithaca, NY) for Sanger sequencing. The sequencing data were analyzed using NCBI Nucleotide Blast homology search to determine the species of those antifungal bacterial isolates [58].

## Optimized production of antifungal compounds

Different growth conditions were tested to optimize antifungal production by the honey isolates. Four media were selected for growth optimization: tryptic soy broth (TSB) (BD Difco, Franklin Lakes, NJ), brain-heart infusion (BHI) (BD Difco, Franklin Lakes, NJ) broth, 1.5% casamino acids (CAA) (BD Difco, Franklin Lakes, NJ) with 0.5% yeast extract (BD Difco, Franklin Lakes, NJ) broth, and potato dextrose broth (PDB) (BD Difco, Franklin Lakes, NJ). Selected growth times were 24 hours or 48 hours, and selected incubation temperature and shaking speed combinations were 37˚C at 250 rpm or 30˚C at 150 rpm. Following the growth of each strain under each condition, the cell-free supernatant was tested for antifungal activity. Cultivated media was first centrifuged at 4˚C, 13000 x *g* for 10 minutes. Supernatant was then filtered through a 0.22 μm polyethersulfone (PES) bottle top filter (250 mL, Celltreat, Pepperell, MA). The cell-free filtrate was tested for antifungal activity using a well diffusion overlay inhibition assay. Wells were made on 25 mL PDA plates using the wide end of sterile 1000 μL pipette tips (diameter: 8.8 mm). A total volume of 600 μL filtrate was added to each well and dried in a biosafety cabinet. Fungal spores were suspended and mixed with 10 mL 0.75% soft TSA and poured onto PDA plates. Plates were incubated at ambient temperature for 48–72 hours, until the complete growth of fungi or the inhibition zone could be visualized. Clear inhibition zones were observed and recorded.

## Purification of antifungal proteinaceous compounds

Two bacterial isolates WRB-ZX-001 and WRB-ZX-002 that showed the ability to excrete antifungal compounds into the broth media were selected for purification. Supernatant of the cell culture grown at optimized condition was treated with ammonium sulfate to precipitate proteins. Solid ammonium sulfate was added to the supernatant at 4˚C to reach saturation of 20%, 40%, 60%, 80% and 100%. Ammonium sulfate precipitates of each percentage saturation were collected separately by centrifugation at 13000 x *g*, 4˚C for 20 min and re-dissolved in sterile Milli-Q $H_2O$. Precipitates were tested against fungal indicator strain *A. fumigatus* and fractions showed antifungal activity were further purified by reversed-phase solid phase extraction (SPE) using a C18 sorbent cartridge (Sep-Pak Classic, Waters, Milford, MA) with acetonitrile as solvent. Acetonitrile with gradient concentrations from 0% to 100% with an increment of

10% was added to eluate the antifungal compounds. All fractions were tested against fungal indicator strain *A. fumigatus* through the well diffusion overlay inhibition assay as described before. Antifungal fractions from SPE were purified via high-performance liquid chromatography (HPLC, Agilent 1200 Series Gradient System, Santa Clara, CA). The following HPLC elution condition was used: 0–10 min mobile phase A (0.05% TFA in dH$_2$O); 10–40 min a gradient of 0–100% mobile phase B (0.05% TFA in acetonitrile); and 40–50 min mobile phase B, with a flow rate of 1 mL/min. Fraction collection from HPLC was performed every 1.5 min. The active fractions were re-injected onto HPLC with the same elution condition to confirm its purity. The antifungal activity of HPLC collected fractions was determined by well diffusion overlay inhibition assay as mentioned previously. Antifungal activity units (AU/mL) of active ammonium sulfate precipitate, SPE fractions and HPLC collected fractions, defined as the reciprocal of the highest dilution yielding a clear inhibition zone, were calculated.

## Growth curve and antifungal production

The growth curves of two selected bacterial isolates, WRB-ZX-001 and WRB-ZX-002, and their antifungal production over time were determined. These two isolates were pre-grown in 5 mL BHI broth at 30˚C, 150 rpm for 12 hours. Pre-growth cell culture (500 μL) was inoculated into 50 mL BHI broth. Samples were taken every two hours from 0 h to 96 h for cell density and antifungal activity measurement. The absorbance of the samples was measured at 600 nm using a spectrophotometer (Spectronic 20D+, Thermo Scientific, Waltham, MA); absorbance values were used to plot growth curves for the two isolates. Antifungal activity was tested by well diffusion overlay inhibition assay of sterile-filtered supernatant against fungal indicator strain *A. fumigatus*. Cell-free supernatants were diluted two-fold and antifungal activity units were calculated as the reciprocal of the highest dilution showing a clear inhibition zone. Biological duplicates were performed. Data was analyzed and visualized in R version 4.0.2. R package growthcurver 0.3.0 was used to fit the microbial growth data to a standard form of logistic equation [59].

## Heat stability and protease stability test

To measure the heat stability and protease stability of the antifungal compounds produced by WRB-ZX-001 and WRB-ZX-002, active antifungal fractions of ammonium sulfate precipitate were selected for testing. For heat stability, samples were treated by steam sterilization at 121˚C for 15 min in an autoclave. Antifungal activity was measured by deferred overlay inhibition assay of 10 μL 2-fold diluted heat-treated samples. The protease stability was tested by incubating the samples individually with 100 μg of pronase E (10 mg/mL, Sigma, St. Lois, MO), α-chymotrypsin (25 mg/mL, Sigma, St. Lois, MO), pepsin (20 mg/mL, Sigma, St. Lois, MO), and trypsin (2.5%, Sigma, St. Lois, MO) at 37˚C for 30 min. Antifungal activity was measured by deferred overlay inhibition assay of 10 μL 2-fold diluted protease-treated samples. Antifungal activity units of heat-treated and protease-treated samples were calculated.

## Protein molecular weight determination via mass spectrometry

Active fractions from SPE were analyzed with direct-infusion mass spectrometry (DIMS) to determine the molecular weight of the antifungal compounds. DIMS was performed on a Triversa Nanomate nanospray direct infusion robot (Advion, Ithaca, NY) attached to a Orbitrap Fusion Lumos Mass Spectrometer (Thermo Fisher Scientific, Waltham, MA). Samples were diluted in 50 mM ammonium formate followed by centrifugation prior to direct infusion. Spectra were acquired in positive ion mode with a resolution setting of 500,000 (at *m/z* 200). Active fractions collected from HPLC were analyzed by liquid chromatography-mass

spectrometry (LC-MS) to measure accurate mass of intact protein. Each sample was diluted with 0.1% formic acid and analyzed by LC-MS with a Dionex RSLCnano HPLC coupled to an OrbiTrap Fusion Lumos (Thermo Fisher Scientific, Waltham, MA) mass spectrometer using a 60 min gradient (2–90% acetonitrile). Sample was resolved using a 75 μm x 150 cm PepMap C4 column (Thermo Scientific, Waltham, MA). MS spectra of protein ions of different charge-states were acquired in positive ion mode with a resolution setting of 120,000 (at *m/z* 200) and accurate mass was deconvoluted using Xcalibur (Thermo Scientific, Waltham, MA). DIMS and LC-MS analyses were performed at Donald Danforth Plant Science Center, Proteomics & Mass Spectrometry Facility (St. Louis, MO).

## Whole genome sequencing and genome analysis

Cell pellets from overnight BHI culture of the isolates were treated with lysozyme (20 mg/mL, Millipore Sigma, St. Lois, MO) and RNase A (Qiagen, Germantown, MD). Genomic DNA was extracted using QiaAMP DNA Minikit (Qiagen, Germantown, MD). Library preparation, quality control, and sequencing were conducted by Cornell University Biotechnology Resource Center (Ithaca, NY) using Nextera XT DNA library preparation and indexing kits (Illumina, San Diego, CA). Illumina MiSeq (Illumina, San Diego, CA) was used to obtain 2 × 250 bp paired-end reads. Reads were trimmed using Trimmomatic (version 0.39) and *de novo* assembled with SPAdes (version 3.13.1) using the default k-mer settings for bacterial genome assembly [60, 61]. Scaffolds less than 500 bp were trimmed and assembly quality was assessed using QUAST (version 4.0) [62]. Average genome coverage was determined using BBmap (version 38.45) and SAMtools (version 1.11) [63]. Genome assemblies of *B. amyloliquefaciens* group type strains were downloaded from the National Center for Biotechnology Information (NCBI) assembly database and average nucleotide identity (ANI) analysis of the isolates was conducted via the OrthoANI method using OAT (version 1.40) with BLAST+ (version 2.9.0) [31]. The draft genomes of *B. velezensis* WRB-ZX-001 and WRB-ZX-002 sequenced in this study, the complete genome of *B. subtilis* 168 as an outgroup, and other 42 genomes of *B. amyloliquefaciens* group extracted from NCBI were used to construct a SNP-based phylogeny. The program kSNP v3.0 was used with a k-mer size of 19 as determined by Kchooser [64]. The core SNPs were used to build the maximum likelihood phylogeny in RAxML v8.2.12 under general time-reversible model with gamma distributed sites (GTRGAMMA) and 1000 bootstrap repetitions [65]. The phylogenetic tree was edited in Fig-Tree v1.4.4 and deposited on Figshare (https://doi.org/10.6084/m9.figshare.16688839). The absolute core SNP distance matrix was calculated using Geneious v2020.2.4. Rapid annotation of the genomes was performed using prokka v1.12 [66]. Functional annotation of the predicted proteins was performed with BLAST2GO v1.4.4 [67]. Additionally, genome annotation was performed by the NCBI using the Prokaryotic Genome Annotation Pipeline (PGAP) database [68]. Putative bacteriocin genes were identified using BAGEL4 [69]. Secondary metabolite genome mining pipeline (antiSMASH) was used to identify potential secondary metabolite synthesis gene clusters [70]. Genome alignment between our isolates and the most closely related type strains was performed using BRIG (version 0.95) [71]. Assembled genomes of *B. velezensis* WRB-ZX-001 and WRB-ZX-002 were submitted to Sequence Read Archive (SRA) and GenBank under the BioProject ID PRJNA580475 and PRJNA596478. SRA accession numbers are SRR10397796 and SRR10729003.

## Supporting information

**S1 Fig. LC-MS spectrum for singly charged *m/z* 1043.5 and doubly charged 522.3 of C$_{14}$ iturin A.** Spectrum was extracted from LC-MS for purified antifungal compounds produced

by *Bacillus velezensis* WRB-ZX-001.
(DOCX)

**S1 Table. Food-isolated fungal strains used in this study as indicators (adapted from Snyder, Churey, and Worobo (2019)).**
(DOCX)

**S2 Table. List of publicly available *Bacillus* spp. genome assembly included in this study.**
(DOCX)

**S3 Table. Gene Ontology (GO) annotation and node score distribution for *Bacillus velezensis* WRB-ZX-001 and WRB-ZX-002.**
(DOCX)

**S4 Table. Absolute core genome SNP distance for 43 *Bacillus amyloliquefaciens* group isolates.**
(XLSX)

## Acknowledgments

The authors would like to thank the College of Agriculture and Life Sciences at Cornell University, Biotechnology Resource Center at Cornell Institute of Biotechnology, and the Proteomics & Mass Spectrometry Facility at the Danforth Plant Science Center for their contribution.

## Author Contributions

**Conceptualization:** Zirui Ray Xiong, Randy W. Worobo.

**Data curation:** Zirui Ray Xiong.

**Formal analysis:** Zirui Ray Xiong.

**Funding acquisition:** Randy W. Worobo.

**Investigation:** Zirui Ray Xiong.

**Methodology:** Zirui Ray Xiong, Randy M. Whittal, Abigail B. Snyder.

**Project administration:** Zirui Ray Xiong.

**Resources:** Zirui Ray Xiong, Mario Cobo, Randy M. Whittal, Abigail B. Snyder.

**Software:** Zirui Ray Xiong, Randy M. Whittal.

**Supervision:** Abigail B. Snyder, Randy W. Worobo.

**Validation:** Zirui Ray Xiong.

**Visualization:** Zirui Ray Xiong.

**Writing – original draft:** Zirui Ray Xiong.

**Writing – review & editing:** Mario Cobo, Randy M. Whittal, Abigail B. Snyder, Randy W. Worobo.

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
