## [Decision Letter · Decision Letter 0]

4 Mar 2022

PONE-D-22-02605Purification and characterization of antifungal lipopeptide produced by Bacillus velezensis isolated from raw honeyPLOS ONE

Dear Dr. Zirui Ray Xiong,

Thank you for submitting your manuscript to PLOS ONE. After careful consideration, we feel that it has merit but does not fully meet PLOS ONE’s publication criteria as it currently stands. Therefore, we invite you to submit a revised version of the manuscript that addresses the points raised during the review process.

ACADEMIC EDITOR:Dear Authors,

improve the paper with the suggestion of the Reviewer 1.

We look forward to receiving your revised manuscript.

Kind regards,

Filippo Giarratana

Academic Editor

PLOS ONE

Journal Requirements:

Additional Editor Comments:

Dear Authors,

improve the paper with the suggestion of the Reviewer 1.

Reviewers' comments:

Reviewer's Responses to Questions

**Comments to the Author**

1. Is the manuscript technically sound, and do the data support the conclusions?

Reviewer #1: Yes

Reviewer #2: Yes

2. Has the statistical analysis been performed appropriately and rigorously? 

Reviewer #1: N/A

Reviewer #2: Yes

3. Have the authors made all data underlying the findings in their manuscript fully available?

Reviewer #1: Yes

Reviewer #2: Yes

4. Is the manuscript presented in an intelligible fashion and written in standard English?

Reviewer #1: Yes

Reviewer #2: Yes

5. Review Comments to the Author

Reviewer #1: Introduction

- Line 45: In the medical field, ……… and safe to use. Is not clear, please rewrite this sentence

- Line 53 in the introduction, delete the dot after infection.

- Line 86: replace antioxidant by compounds

- Line 89: the sentence “B. subtilis H215 was isolated from raw honey that was inhibitory to Byssochlamys fulva H25” need to be revised.

- Line 91: the ref “Lee, Churey (16)” must be corrected as the journal instructions.

- Line 99: please change The potential application of these natural products from microbial source by The potential application of these microbial natural products

Results

- 121: The first sentence should be revised

- Line 123: please change this sentence “These 12 isolates were identified by 16S rRNA sequencing; all isolates were Bacillus spp.” by The 16S rDNA sequence of these 12 isolated strains showed a highest identity to that of several B. subtilis spp.

- Line 124: change the sentence “by ……were summarized respectively” in Table 1 and 2.

- Table 1: this table should be placed in material and methods

- As described in table 2, I don’t understand why you have chosen only Co29 and Co30? and not isolates from clove honey like Co5 and Co-6?. These later have better antifungal activity.

- Legend of table 2 should be improved

- m/z in italic

-line 165: “Based on results from previous studies, we presumed that the ions with m/z value of 1043.55 and 1057.57 were C14 and C15 iturin A [M+H]+, respectively: please add references and the formula of the identified compound. Indeed, precise that C14 was at ion with m/z value of 1057.57 was assigned to C15 iturin A [M+H]+. Ion with m/z value of 1079.55 was assigned to C15 iturin A [M+Na].

- C14 or C15? Please clarify

-Last paragraph page 11: add the figure number

-Figure legend (line 182): “was” and not “were”

-Line 209 to 210: sentence should be revised

-line 401-402: sentence should be revised

Materials and methods

- Please add a title “Antifungal assays” before the paragraph in line 448

Additionally, the authors should check their English writing. It should be improved.

Reviewer #2: The paper “Purification and characterization of antifungal lipopeptide produced by Bacillus

velezensis isolated from raw honey” provides interesting results about the isolation, characterization and potential application of iturin A with antifungal effect. The experiments were well defined and the results are clear. Moreover, the application of a genomic approach showed interesting findings on the detection and identification of secondary metabolites. Other studies documented the effect of iturin A, however this article can provide a suitable contribution on this specific topic.

6. PLOS authors have the option to publish the peer review history of their article (what does this mean?). If published, this will include your full peer review and any attached files.

Reviewer #1: No

Reviewer #2: No

---

## [Author Response · Author response to Decision Letter 0]

10 Mar 2022

We thank the editor for the opportunity to submit a revised version of our manuscript “Purification and characterization of antifungal lipopeptide produced by Bacillus velezensis isolated from raw honey”. We appreciate the feedback from the reviewers, and we have revised our manuscript based on the reviewers’ suggestions. We address the specific issues in a detailed response below. 

Reviewer #1: Introduction

- Line 45: In the medical field, ……… and safe to use. Is not clear, please rewrite this sentence

We have re-written the sentence for better clarity.

- Line 53 in the introduction, delete the dot after infection.

We have made the suggested change.

- Line 86: replace antioxidant by compounds

We have made the suggested revision.

- Line 89: the sentence “B. subtilis H215 was isolated from raw honey that was inhibitory to Byssochlamys fulva H25” need to be revised.

We have revised the sentence for better clarity.

- Line 91: the ref “Lee, Churey (16)” must be corrected as the journal instructions.

We appreciate the reviewer’s comment and have corrected the reference as required by the journal. 

- Line 99: please change The potential application of these natural products from microbial source by The potential application of these microbial natural products

We have made the suggested change.

Results

- 121: The first sentence should be revised

We have re-written the sentence for better clarity.

- Line 123: please change this sentence “These 12 isolates were identified by 16S rRNA sequencing; all isolates were Bacillus spp.” by The 16S rDNA sequence of these 12 isolated strains showed a highest identity to that of several B. subtilis spp.

We have re-written the sentence as suggested for better clarity.

- Line 124: change the sentence “by ……were summarized respectively” in Table 1 and 2.

We have re-written the sentence to avoid confusion.

- Table 1: this table should be placed in material and methods

We have rearranged the table order and put the original table 1 in the supporting information.

- As described in table 2, I don’t understand why you have chosen only Co29 and Co30? and not isolates from clove honey like Co5 and Co-6?. These later have better antifungal activity.

We appreciate the reviewer’s comment. As we mentioned in the manuscript (Line 130-132), the isolates Co29 and Co30 were the only two isolates that were able to produce and excrete antifungal compounds to the extracellular environment. Consequently, these two isolates were selected for the following experiments. 

- Legend of table 2 should be improved

We have re-written the legends for better clarity.

- m/z in italic

We have made the suggested change.

-line 165: “Based on results from previous studies, we presumed that the ions with m/z value of 1043.55 and 1057.57 were C14 and C15 iturin A [M+H]+, respectively: please add references and the formula of the identified compound. Indeed, precise that C14 was at ion with m/z value of 1057.57 was assigned to C15 iturin A [M+H]+. Ion with m/z value of 1079.55 was assigned to C15 iturin A [M+Na].

We appreciate the reviewer’s comment and have added two references on Line 164-165. The molecular formula of the identified compound is also included on Line 165-166.

- C14 or C15? Please clarify

We thank the reviewer’s comment, but we think our phrasing is clear. The peak with m/z value of 1043.55 was C14 iturin A [M+H]+; m/z 1057.57 was C15 iturin A [M+H]+; m/z 1079.55 was C15 iturin A [M+Na]+ (Line 165-166, Fig 3 legend, S1 Fig legend).

-Last paragraph page 11: add the figure number

We appreciate the reviewer’s comment, but the paragraph mentioned by the reviewer was a summary of the genome sequencing data and there were no associated figures.

-Figure legend (line 182): “was” and not “were”

We have made the suggested change.

-Line 209 to 210: sentence should be revised

We have re-written the sentence for better clarity.

-line 401-402: sentence should be revised

We have revised the sentence for better clarity.

Materials and methods

- Please add a title “Antifungal assays” before the paragraph in line 448

We have made the suggested change.

Additionally, the authors should check their English writing. It should be improved.

We appreciate the reviewer’s comment. We have read through our manuscript and corrected any grammar or language issues.

---

## [Editor Report · Decision Letter 1]

22 Mar 2022

Purification and characterization of antifungal lipopeptide produced by Bacillus velezensis isolated from raw honey

PONE-D-22-02605R1

Dear Dr. Zirui Ray Xiong,

We’re pleased to inform you that your manuscript has been judged scientifically suitable for publication and will be formally accepted for publication once it meets all outstanding technical requirements.

Kind regards,

Filippo Giarratana

Academic Editor

PLOS ONE

---

## [Editor Report · Acceptance letter]

28 Mar 2022

PONE-D-22-02605R1 

Purification and characterization of antifungal lipopeptide produced by *Bacillus velezensis* isolated from raw honey 

Dear Dr. Xiong:

I'm pleased to inform you that your manuscript has been deemed suitable for publication in PLOS ONE. Congratulations! Your manuscript is now with our production department. 

Kind regards, 

on behalf of

Dr. Filippo Giarratana 

Academic Editor

PLOS ONE